# Characterization and Comparison of Athletic Performance and Change of Direction Deficit Across Youth Futsal Age Groups

**DOI:** 10.3390/jfmk10020103

**Published:** 2025-03-25

**Authors:** João P. Oliveira, Daniel A. Marinho, Tatiana Sampaio, Sílvio Carvalho, Hugo Martins, Jorge E. Morais

**Affiliations:** 1Departamento de Ciências do Desporto, Universidade da Beira Interior, 6201-001 Covilhã, Portugal; jpco-2001@live.com.pt (J.P.O.); dmarinho@ubi.pt (D.A.M.); tatiana_sampaio30@hotmail.com (T.S.); 2Research Centre for Active Living and Wellbeing (LiveWell), Instituto Politécnico de Bragança, 5300-253 Bragança, Portugal; 3Research Center in Sports Sciences, Health Sciences and Human Development (CIDESD), 6201-001 Covilhã, Portugal; 4Associação de Futebol de Bragança, 5300-091 Bragança, Portugal; silvio.carvalho@live.com.pt; 5Departamento de Desporto, Instituto Superior de Ciências Educativas do Douro, 4560-708 Penafiel, Portugal; 6Departamento de Ciências do Desporto, Instituto Politécnico de Bragança, 5300-253 Bragança, Portugal; hugo.xm@ipb.pt

**Keywords:** futsal, maturation, performance, change of direction, agility

## Abstract

**Background/Objectives:** Futsal players need peak strength, functional capacity, speed, and explosive lower-limb power for optimal performance. The aim of this study was to (i) characterize and compare anthropometric characteristics, lower limb strength/power, dynamic balance, linear sprint speed, COD performance, and COD deficit across young futsal age groups and (ii) identify key predictors of COD deficit. **Methods**: Thirty-three male futsal players from three age groups (U13, U15, and U17) were tested. **Results**: All anthropometric variables showed significant group effects with moderate to strong effect sizes, where U17 presented the largest values, followed by U15 and U13. Strength and power variables presented the same trend. The dynamic balance differed significantly across groups. Significant differences between groups with moderate effect sizes were noted in linear sprints (F = 19.54, *p* < 0.001, η^2^ = 0.57), zigzag (F = 19.54, *p* = 0.002, η^2^ = 0.35), and COD deficit (F = 19.54, *p* < 0.001, η^2^ = 0.46). Post hoc tests revealed that U13 and U15 outperformed U17 in the COD deficit (*p* < 0.002). The COD deficit showed a quadratic relationship with age, initially improving but later declining in older players (*p* < 0.001). The body mass and the posteromedial relative difference predicted the COD deficit, with the coefficient of determination (R^2^) explaining 39% of the COD variance. **Conclusions**: Coaches and practitioners should utilize COD drills to target various movement patterns and account for pubertal timing, as growth can impact performance.

## 1. Introduction

Futsal is a more dynamic and fast-paced version of football that has become extremely popular all over the world [1]. Success in futsal requires a multi-faceted approach encompassing physical [2], physiological [3], tactical [4], technical [5], and psychological [6] skills, with rapid decision making being crucial. Studies have explored the assessment of physical fitness traits in futsal and their impact on performance [7,8]. Research shows that elite futsal players engage in an average of 22 high-intensity sprints per match, spending 25–35% of the total match time at speeds exceeding 18 km/h [7]. Additionally, semi-professional futsal players typically perform sprints lasting 2–4 s, emphasizing the need for explosive speed and rapid recovery [8]. These traits are fundamental to performance and crucial for identifying future talents [9]. However, Mendes et al. [9] also emphasized that anthropometric and physiological characteristics should not be overvalued, especially during early development, given the complex interactions among tactical, technical, psychological, maturational, and environmental factors inherent in talent identification processes.

Due to the smaller number of players (five per team) on a reduced court area, futsal matches involve intermittent high-intensity efforts characterized by frequent short-distance sprints, typically less than 10 m [10]. This high intensity is strongly associated with the demands of smaller relative playing areas [11]. Additionally, players execute abrupt accelerations and decelerations [12] and numerous changes of direction predominantly involving the lower limbs [10,13]. These rapid movements are particularly critical during offensive dribbling, defensive positioning, and transitions between attack and defense [10,12]. Such physical demands require specific field-based tests to evaluate the players’ abilities, allowing coaches and sports scientists to profile athletes and better understand their physical condition, strengths, and weaknesses [14]. Futsal players require high peak strength, functional capacity, speed, and explosive power in their lower limbs to enhance their on-court performance [13]. Linear sprints and change of direction tests effectively assess these qualities [15,16]. Additionally, tests like squat jumps (SJ), countermovement jumps (CMJ), and the reactive strength index (RSI) measure the players’ lower limb strength and power [17]. Regarding the lower trunk’s dynamic balance, the Y-balance test has proven to be reliable in young players [18]. Recent research suggests that linear sprint, jumping performance, and dynamic balance are key factors influencing the change of direction performance [19]. Given the importance of these physical capacities in the change of direction (COD) ability, there is a lack of understanding of how they differ across young age groups. A study by Ayarra and co-workers [2] investigated differences in physical performance among futsal players. Still, they only focused on mostly adult athletes of different competitive levels, including Second Division B, Third Division, and junior players. Interestingly, the study found no significant differences between competitive levels (Second Division B: 5.41 ± 0.26 s; Third Division: 5.48 ± 0.32 s; *p* = 0.68), contradicting their initial hypothesis. Similarly, countermovement jump (CMJ) performance remained comparable across groups (Second Division B: 43.6 ± 5.6 cm; Third Division: 45.8 ± 4.3 cm; Junior: 43.4 ± 4.1 cm), reinforcing the idea that once players reach adulthood, physical differences become less pronounced [2]. One could argue that the sample consisted solely of young adults, who were likely already past their major growth spurts and relatively similar in terms of physical maturity. This highlights the importance of considering young age groups where athletes have not reached maturity. Physiological changes associated with growth spurts occur during puberty [20], perhaps leading to deficits in balance and motor coordination due to disproportional growth of the lower limbs compared to the upper limbs and trunk [21]. Therefore, understanding how age impacts the COD deficit—defined as the additional time required to perform a directional change compared to a linear sprint over an identical distance [22]—is crucial for adjusting training programs, maximizing performance, and reducing the risk of injury.

Recent research has demonstrated that COD performance and deficit vary significantly across sports, sexes, and competitive levels, with stronger and faster players often exhibiting higher COD deficits [23,24]. For example, female athletes from rugby and handball demonstrated distinct COD profiles due to differing neuromuscular demands in their respective sports [24]. However, the interplay between COD ability, physical fitness, and maturation remains underexplored in younger populations, especially in futsal. This study seeks to address this gap by focusing on the predictors of COD deficit across young futsal age groups.

The ability to perform COD tasks is crucial in futsal, notably during frequent dribbling sequences, one-on-one defensive duels, and swift transitions between offensive and defensive roles [25]. Recent studies have highlighted the COD deficit as a critical performance metric for assessing an athlete’s efficiency in changing direction relative to their linear sprint capacity [23,26]. The COD deficit quantifies the additional time required to perform a COD task compared to a linear sprint over the same distance, offering insight into movement efficiency [27]. This measure allows coaches to evaluate an athlete’s ability to decelerate, reorient, and reaccelerate effectively, which are essential for futsal’s dynamic demands. Studies have shown that athletes with greater strength and power capacities often exhibit higher COD deficits, suggesting a trade-off between sprinting speed and COD efficiency [24]. Indeed, athletes with higher levels of isometric, eccentric, and concentric muscular strength demonstrate superior abilities to absorb and generate propulsive forces during rapid directional changes, enhancing their COD efficiency [28]. Such muscular capacities thus likely contribute significantly to COD performance and might influence training strategies to optimize athletes’ directional change capabilities.

The 20 m sprint and zigzag tests were chosen in this study due to their established relevance in assessing linear and multidirectional speed [23]. The zigzag test specifically captures COD ability by requiring players to decelerate and accelerate rapidly, mimicking game-specific scenarios [15,29]. Using these tests together provides a comprehensive assessment of the players’ speed-related performance while isolating the efficiency of COD maneuvers from pure sprinting capacity.

Older age groups typically present bigger anthropometrics, greater strength/power in the lower limbs, and faster linear sprint times than younger age groups [30,31], as their physical aptitudes improve with maturation [30,31]. Consequently, it can be argued that these interrelated physical parameters are representative of a futsal player’s performance as they contribute to the major game demands, such as offense, defense, and rapid transitions between the two [13]. While previous research has examined these physical traits in older age categories [25,32] and competitive levels [2], a gap still exists in our understanding of how these factors interact with age to influence specific aspects of performance in younger players. No current research explores how anthropometric factors, lower limb strength and power, and dynamic balance interact with age to influence COD deficit in this population. This research will hopefully assist in filling this gap in the literature by providing insights into the age-related differences in athletic performance among young futsal players. Understanding these differences is vital for coaches to design effective, age-appropriate training programs that enhance performance and cater to the developmental stages of the athletes.

Therefore, the aims of this study were to (i) characterize and compare anthropometric characteristics, lower limbs’ strength and power, dynamic balance, linear sprint speed, COD performance, and COD deficit across young futsal age groups and (ii) identify key predictors of the COD deficit in this population. It was hypothesized that older players would demonstrate superior anthropometric and physical performance metrics, with an associated improvement in the COD deficit due to maturation and physical fitness development.

## 2. Materials and Methods

### 2.1. Participants

The participants consisted of 33 young male futsal players recruited from their respective regional squads. They were chosen from multiple local clubs based on their superior performance, thus forming representative regional teams. Athletes were grouped by age into three categories: U13 (n = 10; mean age: 12.0 ± 0.55 years), U15 (n = 12; mean age: 14.2 ± 0.41 years), and U17 (n = 11; mean age: 16.2 ± 0.67 years). All three age groups had two training blocks per week, with each block lasting between one and a half and two hours. They were evaluated immediately before their major national competition and were considered Tier 2 athletes [33]. To be included in the measurements, players had to be completely free of pain at the time of the study and training regularly. If someone was receiving medical attention at the time or indicated any pain during the Y-balance test (please report to the methods section), they were excluded from the study. Parents or guardians and players themselves signed an informed consent form. All procedures were performed according to the Declaration of Helsinki regarding human research, and the Polytechnic Ethics Board approved the research (No. 127/2023). Ethical approval date: 4 January 2023.

### 2.2. Research Design

This study characterized and compared a set of anthropometric, lower limbs’ strength and power, dynamic balance, linear sprints, COD test and COD deficit between young futsal age groups. A cross-sectional analysis of young futsal players was conducted. The players were split into three groups according to their chronological age primarily due to practical and organizational constraints related to competitive categories commonly used in futsal (U13, U15, and U17). Nevertheless, maturity offset and peak height velocity (PHV) were assessed to account for maturational effects. Figure 1 presents a visual illustration of the applied tests. The data collection took place on three separate days for each test. On the first day, anthropometric measurements and maturity offset calculations were performed. On the second day, lower limb strength and power tests were conducted, followed by dynamic balance assessments. The third day was dedicated to linear sprints and COD tests. All tests were administered by experienced sports scientists and coaches. Each testing session was scheduled at the time of their training sessions, and players were instructed to follow their regular diet and avoid strenuous activities the day before testing.

### 2.3. Data Collection

#### 2.3.1. Anthropometrics and Maturity Offset

Body mass (in kg) was measured on an electronic scale (MC 780-P, Tanita, Tokyo, Japan) with minimal clothing. Height and sitting height (in cm) were measured using an electronic stadiometer (Seca 242, Seca, Hamburg, Germany). The maturity offset (MO, in years) and peak height velocity (PHV, in years) were calculated as suggested elsewhere [34]. The former represents the years an athlete is away from peak height velocity. If the offset is negative, it means that the athlete has not yet reached peak height velocity. A positive offset indicates that the peak height velocity has already occurred.

#### 2.3.2. Lower Limbs’ Strength and Power

Before data collection, players performed a standardized warm-up based on muscle activation monitored by their coach. Afterward, they became familiar with the jumping tests’ protocols by performing each test with very little effort to understand the biomechanics of each test and ensure the correct technique. The squat jump (SJ, measured in cm), countermovement jump (CMJ, measured in cm), and reactive strength index (RSI, measured in m/s) were utilized to assess lower limb strength and power. Each player performed three attempts for each jump type, with 30 s of rest between attempts of the same jump type and three minutes between different jump tests to minimize fatigue [35]. For SJ, players were instructed to jump vertically from a semi-squat position (approximately 90° knee flexion) without countermovement [36]. For the CMJ, participants performed a rapid downward countermovement (knee flexion to approximately 90°), immediately followed by a maximal vertical jump [36]. For the RSI, athletes executed a drop jump from a box height of 45 cm, with RSI calculated as the ratio between flight time (s) and ground contact time (s) [36]. Detailed test validity and reliability descriptions have been previously reported [35,36]. The best trial was used for further analysis [36]. All tests were measured with an Optojump system (Microgate, Bolzano, Italy) with the bars separated by 1 m [37]. The validity and reliability of this equipment have already been confirmed [37]. Detailed protocols for each test can be found elsewhere [35,36].

#### 2.3.3. Dynamic Balance of the Lower Trunk

The dynamic balance of the lower trunk was assessed using the Y-balance test [18]. Before testing, participants were familiarized with the Y-balance test protocol and practiced until they felt comfortable performing the required movements. The composite score (CS, in %) was generated by averaging and multiplying the sum of the three normalized reach distances by 100. Additionally, the absolute (in cm) and relative (in %) reach differences between lower limbs were calculated to evaluate reach symmetry. Deeper insights about this test can be consulted in other works [18,38]. For qualitative analysis, it has been reported that CSs less than 89% and symmetries greater than 4 cm are more likely to promote injuries [38]. Additional information regarding the Y-Balance protocol can be found elsewhere [36].

#### 2.3.4. Linear Sprint and Change of Direction Tests

Before data collection, players performed a standardized warm-up based on muscle activation monitored by their coach. The linear sprint (partials taken at 5, 10, and 20 m) [39] and zig-zag [29] tests were chosen as performance variables and collected on a hard surface indoor court. The 20 m sprint test and zigzag test were selected due to their established reliability and relevance in assessing linear sprint speed and change of direction (COD) ability in team sports, particularly futsal [15,29]. These tests closely replicate futsal’s specific physical demands, characterized by short linear sprints, frequent accelerations and decelerations, and repeated directional changes. Given futsal’s smaller court dimensions and frequent game interruptions, the 20 m sprint test is more appropriate than longer distances (e.g., 30 or 40 m), ensuring ecological validity and alignment with the sport-specific performance requirements. The zigzag test, with its pre-planned directional changes, mimics futsal-specific movements, while the 20 m sprint isolates linear speed, allowing the calculation of the COD deficit—a measure of directional change efficiency [27,29]. Participants performed a 20 m linear sprint test, which consisted of running this distance in a straight line in the shortest time. The players were encouraged to run at maximum speed for another 5 m after the 20 m mark to ensure that the distance was covered in the shortest possible time. Four sets of gates were placed at the starting line, 5 m, 10 m, and 20 m marks to retrieve information about each section. The zig-zag test (i.e., a zigzag course with four 5 m sections spaced at 100° angles—20 m total distance) is considered a COD test based on the aspects of agility that need to be performed [40]. This test was selected because it tested agility in terms of acceleration, deceleration, and balance management [40]. Its relative simplicity and subject familiarity also suggested that learning effects would be minor [41]. Two sets of gates were placed at the starting line and the finish line. Figure 1 shows the schematics of the two tests. Participants performed each test three times at maximum speed, where players walked back to the starting line, actively resting, while the others performed the test, allowing for a rest period never less than 1–2 min between trials [39]. Subsequently, the fastest time was used for further analysis. All tests were timed with Microgate Witty photocells (Microgate, Bolzano, Italy). They were activated when crossed. The players were given directions to initiate their attempts 0.3 m before the first photocell, and the timer started at their very first movement after crossing the photocells. They were advised to start whenever they felt ready to ensure a quicker and more consistent start [42]. The COD deficit was calculated as the time gap between the 20 m linear sprint time and the zigzag test time, as suggested in the literature [29,41].

### 2.4. Statistical Analyses

All analyses were conducted using IBM SPSS software (Version 29; IBM Corp., Armonk, NY, USA). The normality assumption was analyzed with the Shapiro–Wilk test, which revealed a normal distribution. Homogeneity of variance was verified using Levene’s test. Cohen’s d was calculated with 95% confidence intervals for precise interpretation of effect sizes. The mean plus one standard deviation was calculated as descriptive statistics. The level of significance was set at α = 0.05. One-way ANOVA was used to analyze differences between age groups (U13 vs. U15 vs. U17). Total eta square (η^2^) was selected as the effect size index representing the proportion of total variance attributed to each factor. It was interpreted as (i) without effect if 0 < η^2^ < 0.04, (ii) minimum if 0.04 < η^2^ < 0.25, (iii) moderate if 0.25 < η^2^ < 0.64, and (iv) strong if η^2^ > 0.64 [43]. Whenever appropriate, the Bonferroni correction was used to verify the differences between age groups (*p* < 0.017). Cohen’s d was used to estimate the standardized effect sizes and interpreted as (i) trivial if 0 ≤ d < 0.20, (ii) small if 0.20 ≤ d < 0.60, (iii) moderate if 0.60 ≤ d < 1.20, (iv) large if 1.20 ≤ d < 2.00, (v) very large if 2.00 ≤ d < 4.00, and (vi) nearly distinct if d ≥ 4.00 [44]. Curve fitting was used to understand the relationship between COD deficit and age. The coefficient of determination (R^2^) was used to understand the magnitude of the relationship. Qualitatively, this was defined as very weak if R^2^ < 0.04, weak if 0.04 ≤ R^2^ < 0.16, moderate if 0.16 ≤ R^2^ < 0.49, high if 0.49 ≤ R^2^ < 0.81, and very high if 0.81 ≤ R^2^ <1.0. Simple linear regression (backward method) was used to test the COD deficit predictors, and the R^2^ was used to understand the model variance.

## 3. Results

Table 1 presents the players’ descriptive statistics per age group. The older age group (U17) presented bigger anthropometrics, followed by the U15 and U13, respectively. The lower limb strength and power presented the same tendency, where the U17 age group presented better scores, followed by U15 and U13, respectively (Table 1). Regarding the dynamic balance, an opposite trend was noted. That is, the U13 group presented better scores in the CS (both limbs), followed by U15 and U17, respectively. Additionally, the U17 group presented CS scores (both limbs) below the 89% cut-off value, indicating a risk of injury. In the linear sprint, the U17 group was faster than the U15 and U13, respectively. As for the zigzag test (COD), the fastest age group was U15, followed by U17 and U13, respectively (Table 1).

Table 1 also presents the differences between groups. All anthropometric variables presented a significant group effect with moderate to strong effect sizes. The U17 group exhibited the largest anthropometric values, followed by U15 and U13. Strength and power variables followed a similar trend, with U17 outperforming U15 and U13 in all tests (*p* < 0.05), where the CMJ had the greatest difference between groups (F = 9.72, *p* < 0.001, η^2^ = 0.43). As for the dynamic balance, both CSs presented significant differences between groups (where the right one was greater than the left one). However, non-significant differences were noted by all reaches in both limbs. Linear sprints, zigzag, and the COD deficit presented significant differences between groups with moderate effect sizes (Table 1).

Table 2, Table 3 and Table 4 present pairwise comparisons between age groups (U13 vs. U15, U13 vs. U17, and U15 vs. U17, respectively). Anthropometric differences were observed, with older athletes (U15 and U17) presenting significantly greater body mass and height compared to younger athletes (U13). Regarding strength and power, significant differences favoring older athletes (U17 over U13) were observed in CMJ, SJ, and relative power. The RSI showed no significant differences across age groups. For dynamic balance, only the right-limb composite score (CS right) significantly differed, with the U17 group outperforming the U13 group (mean diff. = 9.957, 95% CI = 4.16 to 15.75, *p* < 0.001, d = 2.59). Linear sprint performance did not significantly differ between U15 and U17 age groups. In the zigzag test, the U15 group was significantly faster than the U13 group. Finally, the COD deficit was significantly lower (better performance) in the U17 group compared to the younger age groups (U13 and U15) (Table 2, Table 3 and Table 4).

Figure 2 presents the association between the COD deficit and age, which was the variable presenting the greatest correlation with the COD deficit (r = −0.533, *p* = 0.001). This indicates that older players are more likely to present smaller COD deficits. However, curve fitting indicated a quadratic association (R^2^ = 0.34, *p* = 0.002). This indicates that the COD deficit tended to decrease with age (best performances) but was followed by an increase with age (poorest performances).

The simple linear regression retained as significant predictors the body mass (β = 0.009, *p* = 0.006) and the posteromedial relative difference (β = 0.031, *p* = 0.013) of the Y-balance test. Thus, the prediction equation is as follows:(1)COD deficit=1.497+0.009·BM+0.031·posteromedial_rel_diff
where the COD deficit is the change of direction deficit (in s), BM is the body mass (in kg), and posteromedial_rel_diff is the posteromedial relative difference obtained in the Y-balance test (in %).

## 4. Discussion

The aims of this study were to (i) characterize and compare anthropometric characteristics, lower limbs’ strength and power, dynamic balance, linear sprint speed, COD performance, and COD deficit across young futsal age groups and (ii) identify key predictors of the COD deficit in this population. The main findings indicate that the U17 age-group was significantly bigger, with greater lower limb strength and power, and faster in linear sprinting, followed by the U15 and U13 groups, respectively. On the other hand, inverse findings were noted for the dynamic balance, where the U13 age group presented better CS scores (both limbs), followed by the U15 and U17 groups, respectively. In the COD test (i.e., zigzag), the fastest age group was U15, followed by U17 and U13. As for the COD deficit, U15 was also the age group with the best scores, followed by U13 and U17. The COD deficit presented a quadratic relationship with age. That is, it initially decreased (improved) from U13 to U15, likely associated with improved coordination and COD-specific training adaptations, coinciding with the approach toward PHV. However, a consistent increase (poorer performance) from U15 to U17 was observed, possibly explained by maturational effects such as increased body mass and transient motor coordination deficits immediately following PHV. Finally, the COD deficit was predicted by the body mass and the posteromedial relative difference (where greater values led to a poorer COD deficit).

These findings partially support our initial hypothesis. That is, older players (U17) exhibited bigger anthropometrics, greater strength/power, and faster linear sprint times compared to younger age groups (U15 and U13). This aligns with expectations and suggests a maturational effect on these physical qualities relevant to futsal performance. Previous research in youth futsal has shown a positive association between chronological age and key physical traits such as anthropometrics, lower limb power, and linear sprint speed [30,31]. In our study, maturity offset and PHV data confirmed that U17 athletes mainly were past their PHV, while the U15 athletes experienced or approached their peak height velocity, thus partly explaining the observed anthropometric and performance differences. Similarly, [45] demonstrated comparable trends in football, which shares similar physical demands with futsal [46], requiring a strong foundation in anthropometrics, lower limb power, and linear speed. However, while our findings align with the general expectation that maturation enhances physical performance, they also reveal a potential trade-off: stronger and faster athletes may exhibit higher COD deficits, reflecting reduced efficiency in directional changes relative to their sprint capacity [23,24]. Interestingly, unlike studies on adult athletes from various team sports [23], the younger futsal players in this study showed greater variability in COD performance, likely due to differences in maturation and training history. These results underscore the need for age-specific COD training protocols that account for developmental stages and individual variability in young athletes.

On the other hand, an unexpected finding with the dynamic balance assessment was verified. The U13 group exhibited better scores compared to older groups (U15 and U17). This could be attributed to physiological changes associated with growth spurts [20]. During puberty, the lower limbs experience a more rapid linear increase than the upper limbs and trunk [21,47]. This disproportional growth can lead to transient deficits in balance and motor coordination due to body instability, potentially affecting performance in dynamic activities [21,47]. This instability might be linked to altered stride frequency during running caused by the temporary mismatch between leg length and core strength [21,47]. Given the typical timing of puberty, the U13 players in our study are less likely to be affected by these growth-related changes than the U15 and U17 groups. The older age groups would have already begun experiencing these disproportional growth spurts, potentially contributing to the lower dynamic balance scores observed. While the research on the specific impact of growth spurts on stride mechanics in adolescents is limited [30], this temporary imbalance could explain our study’s lower dynamic balance scores observed in the older age groups (U15 and U17).

Another factor that might contribute to the unexpected finding of lower dynamic balance scores in the U15 and U17 groups compared to U13 is the influence of the musculoskeletal system during puberty. As adolescents progress through puberty, they experience significant increases in muscle mass [21,48,49]. This muscle growth is often accompanied by increased muscle-tendon stiffness [50]. Studies suggest that increased stiffness can hinder the body’s ability to make rapid adjustments required during dynamic activities [21], such as the Y-balance test. This concept aligns with the observations by Laffaye et al. [51], who reported an increase in lower limb stiffness with increasing chronological age in young athletes. They attributed this finding to more elastic tissue (tendons, fascia) associated with increased muscle mass [51]. Additionally, the structural properties of collagen fibers within tendons are thought to contribute to stiffness [52,53]. These fibers stiffen under rapid stretching, potentially limiting the responsiveness needed for optimal dynamic balance performance [52,53]. While the specific impact of muscle tendon stiffness on dynamic balance in young athletes remains a topic for further investigation, it presents a plausible explanation for the observed results in our study, particularly for the older age groups (U15 and U17).

The analysis of the COD deficit revealed an interesting trend. Initially, the COD deficit decreased with age, with the U15 group exhibiting the lowest deficit. This finding aligns with expectations, as young athletes typically experience improved coordination, technique development specific to futsal, and the effects of training programs designed to enhance COD ability [54,55]. However, this trend did not continue in the U17 age group, where the COD deficit unexpectedly increased. A possibility relates to the physiological changes associated with growth spurts during puberty. As discussed earlier, growth spurts can lead to transient deficits in balance and motor coordination due to temporary body instability caused by disproportional growth of the lower limbs compared to the upper limbs and trunk [20]. This instability can manifest as altered stride frequency during running and potentially affect COD performance through similar mechanisms that may have impacted dynamic balance in this age group [47].

Information about the effect of age on the COD deficit in youth futsal is scarce [31]. However, studies in football, a closely related sport, can provide a general framework for interpreting our findings. There are similarities between our results and football research. Nevertheless, the unexpected U-shaped relationship between age and COD deficit (improvement followed by decline) might differ from football research, where COD ability is generally thought to improve linearly with age and maturation [56]. This difference could be attributed to the specific demands of futsal compared to football. Futsal places a greater emphasis on quick changes of direction in a confined space, potentially requiring more refined technique and decision making compared to the larger playing field of football [13,57,58], or as mentioned earlier, the U17 group might be experiencing negative temporary effects of growth spurts on coordination until athletes adapt to their changing body composition.

The linear regression analysis identified body mass and the posteromedial relative difference as significant predictors of COD deficit in young futsal players. The finding that increased body mass is associated with a greater COD deficit aligns with existing literature on the importance of body composition in athletic performance [59,60,61]. Heavier players, particularly those with a higher proportion of fat mass, often exhibit reduced agility and slower change of direction speeds due to the increased effort required to move their body mass rapidly [28,62]. Conversely, a higher percentage of lean mass has been shown to enhance performance in activities requiring speed and agility, contributing to greater muscle power and efficiency [28,62]. This suggests that training programs aimed at optimizing body composition by increasing lean mass and reducing fat mass could improve COD ability in futsal players.

Additionally, the posteromedial relative difference, which measures asymmetry in dynamic balance, was found to predict the COD deficit. This is logical as an effective change of direction involves coordinated movements to both sides [63]. An imbalance in this ability might indicate that an athlete is more proficient in moving in one direction than the other, thereby impairing overall agility and increasing the time to change direction. Research indicates that dynamic balance, assessed using tests like the Y Balance Test, is essential for effective COD [64]. Significant associations between dynamic balance and COD performance emphasize the importance of balance training in improving agility [64]. Previous studies have similarly highlighted the impact of bilateral asymmetry on performance metrics, emphasizing the need for balanced development in athletic training [65,66]. Addressing these imbalances through targeted strength and conditioning programs, as well as incorporating exercises that promote bilateral symmetry, could help in reducing COD deficits and enhancing overall performance. It is suggested that athletes with better balance can distribute force more evenly during directional changes, leading to improved performance and reduced injury risk [66]. Future research should continue to explore the role of these and other potential predictors to better understand and improve COD ability in young athletes.

Overall, it was noted that older players had bigger anthropometrics, greater lower limb strength, and were faster at linear sprinting. However, in the case of dynamic balance, the trend was inverted. The older players presented the worst values of dynamic balance, followed by the U15 and U13 groups (reduced with age). Therefore, coaches and practitioners should regularly monitor their players’ dynamic balance and employ dedicated training designs to minimize performance impairment. Regarding the COD deficit, initially, it was noted to decrease with age (best performances). Afterward, a consistent increase (poorest performances) was noted, indicating that players started to lose their ability to change direction effectively in the older age groups. Body mass and posteromedial relative differences emerged as key predictors of COD deficit in young futsal players, emphasizing the importance of a balanced body composition and dynamic balance in optimizing athletic performance.

This study contributes to the limited body of research on COD deficit in youth futsal players by identifying age-specific differences in performance and their underlying predictors. Unlike previous studies focused on adult athletes or football players, our findings highlight the developmental challenges faced by young players, particularly the impact of growth spurts on dynamic balance and COD efficiency. By emphasizing the role of body composition and balance asymmetries, this research provides actionable guidance for coaches and practitioners. Targeted, age-appropriate interventions are crucial to enhancing performance and minimizing injury risks in futsal, a sport where COD efficiency is critical to success. Coaches should consider incorporating a wider range of COD drills that target various movement patterns relevant to futsal and pay attention to the timing of puberty, as growth-related changes might interfere with performance. Further longitudinal research is needed to understand better the potential factors contributing to this decline in COD deficit performance.

As main limitations, it can be considered that (i) only one test of COD was applied to understand the COD deficit. Futsal requires rapid reactive changes of direction involving distinct neuromuscular mechanisms, which were not directly assessed in this study. Therefore, future research should incorporate reactive COD tests to obtain deeper insights into futsal-specific performance demands and to ensure broader applicability of the results. (ii) Cross-sectional design: the nature of this study captures differences between age groups at a single point in time. Longitudinal studies tracking the same cohort over their developmental stages are needed to fully understand the progression of COD performance and its underlying predictors in young futsal players. (iii) The sample size: an a priori power analysis using G*Power [67] revealed that 66 participants were required to detect a large effect size (f^2^ = 0.40) with 80% power (α= 0.05) for an “ANOVA: Fixed effects, omnibus, one-way” statistical test. Nonetheless, it must be mentioned that the sample was composed of the best players from several futsal clubs that play in the district and compete at the national level and (iv) the fact that the players’ athletic history was not controlled. Therefore, future research could employ a broader range of COD tests or a combination of tests that better reflect the demands of futsal playing, performing the same type of analysis with a larger sample size, with other age groups, and with better knowledge of the players’ athletic history.

## 5. Conclusions

Older players are significantly bigger, stronger, and faster at linear sprinting. An unexpected finding with dynamic balance was noted, where the youngest age group displayed the best scores. This might be attributed to growth spurt-related instability in older players. The COD deficit initially decreased with age, suggesting improvement. However, the increase in the U17 group warrants further investigation. The COD deficit was predicted by the body mass and posteromedial relative difference obtained during the Y-balance test. Greater values of these variables led to a poorer COD deficit. Therefore, coaches and practitioners should utilize COD drills to target various movement patterns and account for pubertal timing, as growth can impact performance.

## Figures and Tables

**Figure 1 jfmk-10-00103-f001:**
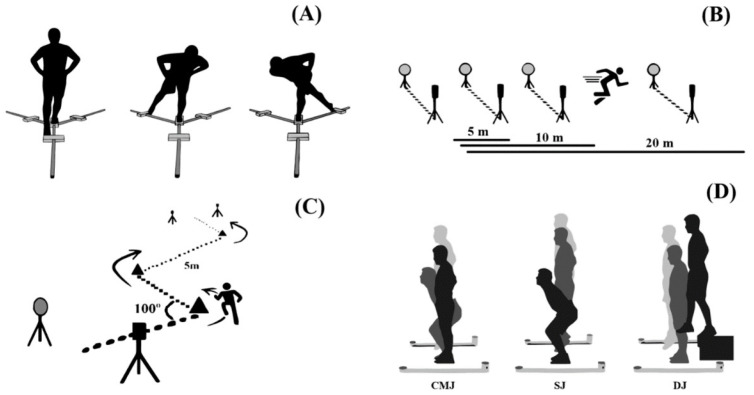
Visual presentation of the applied tests. Panel (**A**): Y-balance test; Panel (**B**): 20 m linear sprint test; Panel (**C**): 20 m zigzag agility test; Panel (**D**): jump tests (CMJ—countermovement jump; SJ—squat jump; DJ—drop jump).

**Figure 2 jfmk-10-00103-f002:**
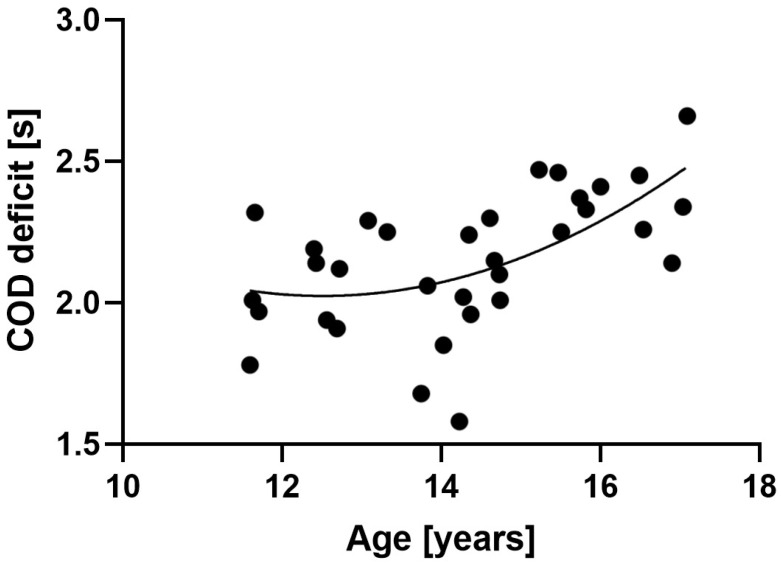
Association between the COD deficit and age. COD—change of direction.

**Table 1 jfmk-10-00103-t001:** Descriptive statistics (mean and standard deviation—SD) of all variables by age group. This table also presents the one-way ANOVA to identify the differences between groups.

	U13 N = 10		U15 N = 12		U17 N = 11			
	Mean	SD	Mean	SD	Mean	SD	F-Ratio (*p*-Value)	η^2^
	**Demographics and Anthropometrics**							
**Decimal age [years]**	12.24	0.55	14.24	0.44	16.17	0.67	128.88 (0.001)	0.90
**Years of practice [years]**	3.80	1.14	4.92	1.24	5.73	1.49	5.782 (0.008)	0.28
**Body mass [kg]**	42.92	7.74	57.53	9.22	65.60	11.03	14.22 (0.001)	0.50
**Height [cm]**	156.89	8.96	170.08	5.57	174.73	4.86	19.74 (<0.001)	0.58
**Sitting height [cm]**	78.56	4.45	82.00	8.73	86.18	3.87	10.22 (<0.001)	0.41
**Maturity offset [years]**	−4.07	0.36	−0.13	1.22	1.26	0.81	183.94 (<0.001)	0.93
**PHV [years]**	16.26	0.51	14.01	0.33	14.90	0.77	38.76 (<0.001)	0.74
	**Strength and Power**							
**CMJ height [cm]**	24.28	3.00	30.84	3.47	32.39	5.84	9.72 (<0.001)	0.43
**SJ height [cm]**	23.22	3.56	29.03	4.33	31.53	5.80	7.60 (0.003)	0.37
**DJ height [cm]**	20.53	3.33	26.20	4.77	24.88	4.07	4.93 (0.015)	0.28
**Power [w/kg]**	23.17	2.60	29.14	4.84	29.47	4.41	6.70 (0.004)	0.34
**RSI [m/s]**	0.68	0.11	0.92	0.22	0.95	0.19	5.85 (0.008)	0.31
	**Dynamic Balance**							
**CS_right_ [%]**	94.63	4.56	89.37	6.49	84.67	2.85	9.72 (<0.001)	0.43
**CS_left_ [%]**	94.12	6.72	89.37	5.54	86.59	3.64	4.47 (0.021)	0.26
	**Anterior differences**							
**Absolute [cm]**	3.65	1.99	2.47	1.45	4.04	1.82	2.08 (0.145)	0.14
**Relative [%]**	4.21	2.53	3.20	3.37	4.49	2.12	0.59 (0.560)	0.04
	**Posterolateral differences**							
**Absolute [cm]**	4.78	3.49	2.53	1.58	4.30	3.34	1.64 (0.214)	0.11
**Relative [%]**	5.63	4.67	2.57	1.67	4.69	3.36	2.05 (0.149)	0.14
	**Posteromedial differences**							
**Absolute [cm]**	3.55	1.88	3.81	3.93	5.75	4.57	1.03 (0.370)	0.07
**Relative [%]**	4.75	1.96	2.91	2.35	6.23	5.05	2.37 (0.114)	0.15
	**Linear sprint**							
**Time 5 m [s]**	1.17	0.08	1.11	0.07	1.03	0.05	11.56 (<0.001)	0.44
**Time 10 m [s]**	2.01	0.11	1.87	0.09	1.78	0.07	16.84 (<0.001)	0.53
**Time 20 m [s]**	3.52	0.20	3.20	0.17	3.09	0.11	19.54 (<0.001)	0.57
	**Agility**							
**Zigzag [s]**	5.59	0.17	5.22	0.28	5.47	0.18	7.99 (0.002)	0.35
**COD deficit [s]**	−2.07	0.17	−2.02	0.22	−2.38	0.14	12.55 (<0.001)	0.46

PHV—peak height velocity; CMJ—countermovement jump; SJ—squat jump; DJ—drop jump; RSI—reactive strength index; CS—composite score; COD—change of direction; η^2^—eta square (effect size index).

**Table 2 jfmk-10-00103-t002:** Pairwise comparison of all variables between U13 and U15 groups. Only significant differences (*p* < 0.017) are presented.

	U13 vs. U15			
	Mean Diff.	95% CI	*p*-Value	d [Descriptor]
	**Demographics and Anthropometrics**			
**Decimal age [years]**	−1.995	−2.60 to −1.39	<0.001	4.05 [nearly distinct]
**Body mass [kg]**	−14.603	−25.27 to −3.93	0.005	1.69 [large]
**Height [cm]**	−13.194	−20.46 to −5.93	<0.001	1.83 [large]
**Sitting height [cm]**	−5.936	−10.31 to −1.57	0.005	3.92 [very large]
**Maturity offset [years]**	−4.256	−4.98 to −3.53	<0.001	7.90 [nearly distinct]
**PHV [years]**	2.250	1.60 to 2.90	<0.001	5.36 [nearly distinct]
	**Strength and Power**			
**CMJ [cm]**	−6.564	−11.22 to −1.90	0.004	2.00 [very large]
**SJ [cm]**	NS			
**DJ [cm]**	−5.667	−10.39 to −0.95	0.015	1.34 [large]
**Power [w/kg]**	−5.967	−10.65 to −1.28	0.009	1.47 [large]
**RSI [m/s]**	NS			
	**Dynamic Balance**			
**CS_right_ [%]**	NS			
**CS_left_ [%]**	NS			
	**Anterior differences**			
**Absolute [cm]**	NS			
**Relative [%]**	NS			
	**Posterolateral differences**			
**Absolute [cm]**	NS			
**Relative [%]**	NS			
	**Posteromedial differences**			
**Absolute [cm]**	NS			
**Relative [%]**	NS			
	**Linear sprint**			
**Time 5 m [s]**	NS			
**Time 10 m [s]**	0.139	0.04 to 0.24	0.004	1.39 [large]
**Time 20 m [s]**	0.319	0.14 to 0.50	<0.001	1.75 [large]
	**Agility**			
**Zigzag [s]**	0.370	0.13 to 0.61	0.002	1.54 [large]
**COD deficit [s]**	NS			

PHV—peak height velocity; CMJ—countermovement jump; SJ—squat jump; DJ—drop jump; RSI—reactive strength index; CS—composite score; COD—change of direction; 95% CI—95% confidence intervals; d—Cohen’s d (effect size index); NS—non-significant.

**Table 3 jfmk-10-00103-t003:** Pairwise comparison of all variables between U13 and U17 groups. Only significant differences (*p* < 0.017) are presented.

	U13 vs. U17			
	Mean Diff.	95% CI	*p*-Value	d [Descriptor]
	**Demographics and Anthropometrics**			
**Decimal age [years]**	−3.918	−4.54 to −3.30	<0.001	6.36 [nearly distinct]
**Body mass [kg]**	−22.678	−33.56 to −11.80	<0.001	2.34 [very large]
**Height [cm]**	−17.838	−25.25 to −10.43	<0.001	2.55 [very large]
**Sitting height [cm]**	−7.626	−12.08 to −3.17	<0.001	4.14 [nearly distinct]
**Maturity offset [years]**	−5.329	−6.07 to −4.59	<0.001	8.19 [nearly distinct]
**PHV [years]**	1.358	0.71 to 2.01	<0.001	2.04 [very large]
	**Strength and Power**			
**CMJ [cm]**	−8.110	−13.25 to −2.97	0.001	1.78 [large]
**SJ [cm]**	−8.303	−13.99 to −2.62	0.003	1.75 [large]
**DJ [cm]**	NS			
**Power [w/kg]**	−6.285	−11.45 to −1.12	0.013	1.77 [large]
**RSI [m/s]**	NS			
	**Dynamic Balance**			
**CS_right_ [%]**	9.957	4.16 to 15.75	<0.001	2.59 [very large]
**CS_left_ [%]**	NS			
	**Anterior differences**			
**Absolute [cm]**	NS			
**Relative [%]**	NS			
	**Posterolateral differences**			
**Absolute [cm]**	NS			
**Relative [%]**	NS			
	**Posteromedial differences**			
**Absolute [cm]**	NS			
**Relative [%]**	NS			
	**Linear sprint**			
**Time 5 m [s]**	0.142	0.07 to 0.22	<0.001	2.14 [very large]
**Time 10 m [s]**	0.231	0.13 to 0.33	<0.001	2.48 [very large]
**Time 20 m [s]**	0.428	0.25 to 0.61	<0.001	2.69 [very large]
	**Agility**			
**Zigzag [s]**	NS			
**COD deficit [s]**	0.309	0.11 to 0.51	0.002	1.98 [large]

PHV—peak height velocity; CMJ—countermovement jump; SJ—squat jump; DJ—drop jump; RSI—reactive strength index; CS—composite score; COD—change of direction; 95% CI—95% confidence intervals; d—Cohen’s d (effect size index); NS—non-significant.

**Table 4 jfmk-10-00103-t004:** Pairwise comparison of all variables between U15 and U17 groups. Only significant differences (*p* < 0.017) are presented.

	U15 vs. U17			
	Mean Diff.	95% CI	*p*-Value	d [Descriptor]
	**Demographics and Anthropometrics**			
**Decimal age [years]**	−1.924	−2.52 to −1.33	<0.001	3.42 [very large]
**Body mass [kg]**	N.S.			
**Height [cm]**	N.S.			
**Sitting height [cm]**	N.S.			
**Maturity offset [years]**	−1.072	−1.76 to −0.39	0.001	1.48 [large]
**PHV [years]**	−8.918	−1.51 to −0.27	0.003	1.51 [large]
	**Strength and Power**			
**CMJ [cm]**	NS			
**SJ [cm]**	NS			
**DJ [cm]**	NS			
**Power [w/kg]**	NS			
**RSI [m/s]**	NS			
	**Dynamic Balance**			
**CS_right_ [%]**	NS			
**CS_left_ [%]**	NS			
	**Anterior differences**			
**Absolute [cm]**	NS			
**Relative [%]**	NS			
	**Posterolateral differences**			
**Absolute [cm]**	NS			
**Relative [%]**	NS			
	**Posteromedial differences**			
**Absolute [cm]**	NS			
**Relative [%]**	NS			
	**Linear sprint**			
**Time 5 m [s]**	NS			
**Time 10 m [s]**	NS			
**Time 20 m [s]**	NS			
	**Agility**			
**Zigzag [s]**	NS			
**COD deficit [s]**	0.360	0.17 to 0.55	<0.001	1.92 [large]

PHV—peak height velocity; CMJ—countermovement jump; SJ—squat jump; DJ—drop jump; RSI—reactive strength index; CS—composite score; COD—change of direction; 95% CI—95% confidence intervals; d—Cohen’s d (effect size index); NS—non-significant.

## Data Availability

The data presented in this study are available upon request from the corresponding author (the data are not publicly available due to privacy or ethical restrictions).

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
