# Peer review of "Characterization and Comparison of Athletic Performance and Change of Direction Deficit Across Youth Futsal Age Groups"

_jfmk, 2025, doi:10.3390/jfmk10020103_

Round 1

Reviewer 1 Report

Comments and Suggestions for Authors

This study aims to characterize and compare anthropometric data, lower limb power, dynamic balance, linear sprint, COD performance, and COD deficit in athletes from U13, U15, and U17 categories.

I would like to congratulate the authors on their work, which appears to be relevant to the field and is well-structured and elaborated. However, I offer some suggestions for improvement.

Abstract

Well-structured, concise, and contains the necessary information for an adequate summary of the study.

Introduction

  • Line 38 – I suggest the authors remove "The" at the beginning of the first paragraph, leaving only "Futsal is a dynamic and fast-paced variant of football...", as the term "Futsal" does not require a definite article in English. Additionally, in the same paragraph, I suggest replacing "all over the world" with "worldwide" for conciseness and natural flow.
  • Line 39 – The second idea in the second paragraph is extremely interesting, but I suggest the authors move it to the next paragraph (line 45), where they describe the physical demands of futsal. This way, it can be justified not only by field size but also by the number of players, highlighting the importance of relative playing area.
  • Lines 42-44 – I agree with the authors in associating physical abilities with performance, as there is specific literature supporting this connection. However, I recommend caution when linking this to talent identification. To date, I am unaware of any studies that directly associate physical capabilities with talent detection in futsal, which should always be analyzed holistically and as a multifactorial process.
  • Lines 45-46 – Considering the study’s objective, I recommend the authors review this paragraph and provide more details about the physical demands of futsal. Additionally, changes of direction should be described due to their importance in the study. The authors could also contextualize the importance of these variables in gameplay, explaining in which situations players frequently accelerate, decelerate, or change direction. Furthermore, I suggest quantifying some of these metrics in meters/kilometers or frequency to help readers understand the actual demands (this could also help justify the choice of physical tests in the methodology).
  • Line 68 – The authors should clarify the concept of COD deficit the first time it is mentioned.
  • Lines 70-77 – Several terms related to COD are used: COD performance, COD deficit, COD profiles, COD ability. While I understand the need for different terms, I suggest unifying the terminology for consistency.
  • Lines 78-79 – I fully agree with the authors, but I suggest contextualizing the actions related to COD in futsal, such as dribbling or defensive movements, to strengthen the connection with game dynamics.
  • Lines 86-88 – The authors could enhance this idea by mentioning that athletes with higher isometric, eccentric, and concentric strength indices have greater ability to absorb and apply propulsive force during direction changes. (Spiteri et al., 2015 - Mechanical determinants of faster change of direction and agility performance in female basketball athletes).
  • Lines 89-90 – Although the selected tests align with the study’s goals, the referenced study does not include futsal athletes. I suggest reformulating the sentence or selecting a more relevant reference.

Materials and Methods

  • Line 119 – The authors should describe the sample in greater detail. For future study comparisons, it is essential to specify not only the age but also the competitive level of the athletes. While the use of Tier 2 classification is interesting, some readers may not be familiar with it. Are all athletes from the same team? How many athletes are from U13, U15, and U17? Information on competition level, number of participating teams, and team ranking would enhance the description.
  • Line 137 – Why did the authors define groups based on age rather than maturation level?
  • Lines 163-164 – I acknowledge the challenges of data collection in team settings, but do the authors consider that greater prior familiarization with the tests could enhance consistency in the results?
  • Line 165 – In Figure D, the image and caption depict a drop jump. While I understand that the measured variable is RSI, the test name should be consistent. The RSI calculation formula should also be provided. Additionally, the drop height (in cm) must be specified. A detailed movement description is recommended to ensure proper execution of the three jumps.
  • Line 188 – Court or pitch?
  • Lines 186-187 – Can the authors justify the choice of tests for futsal? Why was the 20m sprint test selected instead of 30m or 40m? Regarding COD tests, were other tests such as the V-Cut or T-Test considered?
  • Line 190 – I suggest using more specific references with greater scientific impact to support this statement.
  • Line 198 – A more detailed description of the test organization is needed to improve replicability. Does the test involve a total linear distance of 20m? What is the distance between markers?
  • Line 212 – I recommend replacing "by others" with "in the literature".

Statistical Analysis

  • Maturity offset and PHV can vary significantly, which may bias results when comparing only by age. Did the authors consider using an ANCOVA to adjust age effects based on maturation?
  • Was homogeneity of variance tested? Was Levene’s test conducted?
  • Authors should specify that, in addition to Cohen’s d, 95% confidence intervals were used for effect size precision.

Results

  • I suggest reducing the written text to essential points to avoid duplicating information between tables and text.
  • Tables – To facilitate result analysis, could Tables 2, 3, and 4 be merged into a single table, possibly in landscape format? This could improve visualization and data comparison. Just a suggestion.

Discussion

  • Lines 466-469 – The authors have maturity offset and PHV data, which could enhance discussion depth rather than just mentioning maturation.
  • Lines 474-481 – Could the differences in COD test results between groups be related to the fact that most U17 athletes have already passed PHV, while U15 athletes are currently experiencing it? What anthropometric and developmental implications might be involved?
Comments on the Quality of English Language

Overall, the English language quality is good, but some phrasing that could be improved for better readability. A thorough proofreading or a language revision service is recommended to ensure clarity and consistency throughout the manuscript.

Author Response

Responses to Reviewer 1 Comments
Abstract
Comment 1: "Well-structured, concise, and contains the necessary information for an adequate
summary of the study."
Authors’ answer: We appreciate the reviewer’s positive feedback regarding the abstract.
Introduction
Comment 2: "Line 38 – I suggest the authors remove 'The' at the beginning of the first paragraph,
leaving only 'Futsal is a dynamic and fast-paced variant of football…', as the term 'Futsal' does
not require a definite article in English. Additionally, in the same paragraph, I suggest replacing
'all over the world' with 'worldwide' for conciseness and natural flow."
Authors’ answer: We appreciate the reviewer’s suggestion. The word “The” was retained by
mistake when relocating the information to the journal’s template. The text has been revised as
follows:
Original: "The Futsal is a more dynamic and fast-paced version of football, that has become
extremely popular all over the world [1]."
Revised: "Futsal is a more dynamic and fast-paced version of football, that has become
extremely popular all over the world [1]."
Comment 3: "Line 39 – The second idea in the second paragraph is extremely interesting, but I
suggest the authors move it to the next paragraph (line 45), where they describe the physical
demands of futsal. This way, it can be justified not only by field size but also by the number of
players, highlighting the importance of relative playing area."
Authors’ answer: We appreciate the reviewer’s valuable suggestion. As recommended, we
relocated and integrated the information to the subsequent paragraph (around line 45). We
revised this paragraph to include justification based on the reduced number of players, smaller
court dimensions, and the resulting high-intensity physical demands:
Revised Paragraph: "Due to the smaller number of players (five per team) on a reduced court
area, futsal matches involve intermittent high-intensity efforts characterized by frequent shortdistance sprints, typically less than 10 meters [10]. This high intensity is strongly associated with
the demands of smaller relative playing areas [11]. Additionally, players execute abrupt
accelerations and decelerations [12] and numerous changes of direction predominantly
involving the lower limbs [10,13]. These rapid movements are particularly critical during actions
such as offensive dribbling, defensive positioning, and transitions between attack and defense
[10,12]. Such physical demands require specific field-based tests to evaluate players' abilities,
allowing coaches and sport scientists to profile athletes and better understand their physical
condition, strengths, and weaknesses [14]."
Comment 4: "Lines 42-44 – I agree with the authors in associating physical abilities with
performance, as there is specific literature supporting this connection. However, I recommend
caution when linking this to talent identification. To date, I am unaware of any studies that
directly associate physical capabilities with talent detection in futsal, which should always be
analyzed holistically and as a multifactorial process."
Authors’ answer: We appreciate the reviewer’s insightful suggestion and acknowledge that
caution is indeed necessary when associating physical capabilities with talent identification in
futsal. Mendes et al. (2022) conducted a comprehensive systematic review on talent
identification and development in male futsal, concluding that although anthropometric and
physiological factors—including aerobic power, body composition, strength, and speed—are
positively associated with successful futsal performance, they alone should not be
overemphasized. Particularly in early developmental stages, talent identification processes must
adopt a holistic approach, carefully integrating physical attributes with technical, tactical,
psychological, maturational, and environmental factors. This complex interplay among multiple
dimensions confirms that talent detection in futsal is inherently multifactorial and dynamic.
Therefore, to accurately reflect this important nuance, we revised our manuscript as follows:
Revised text: " These traits are important for identifying future talents [9]. However, Mendes et
al. [9] also emphasized that anthropometric and physiological characteristics should not be
overvalued, especially during early development, given the complex interactions among tactical,
technical, psychological, maturational, and environmental factors inherent in talent
identification processes."
Comment 5: "Lines 45-46 – Considering the study’s objective, I recommend the authors review
this paragraph and provide more details about the physical demands of futsal. Additionally,
changes of direction should be described due to their importance in the study. The authors could
also contextualize the importance of these variables in gameplay, explaining in which situations
players frequently accelerate, decelerate, or change direction. Furthermore, I suggest
quantifying some of these metrics in meters/kilometers or frequency to help readers
understand the actual demands (this could also help justify the choice of physical tests in the
methodology)."
Authors’ answer: We appreciate the reviewer’s insightful feedback. Following the
recommendation, we expanded the description by clearly quantifying key futsal-specific physical
demands (e.g., sprint distances <10 meters), and clarified the critical game contexts involving
frequent accelerations, decelerations, and directional changes (e.g., dribbling, and defensive
positioning). These specific details further justify our selection of physical tests in the
methodology.
See the revised paragraph provided on comment 3.
Comment 6: "Line 68 – The authors should clarify the concept of COD deficit the first time it is
mentioned."
Authors’ answer: We appreciate the reviewer's suggestion. As recommended, we have clearly
defined the concept of COD deficit at its first mention in the manuscript. The revised sentence
now reads: "Therefore, understanding how age impacts COD deficit — defined as the additional
time required to perform a directional change compared to a linear sprint over an identical
distance [25] — is crucial for adjusting training interventions according to athletes'
developmental stages, thereby optimizing their performance."
Comment 7: "Lines 70-77 – Several terms related to COD are used: COD performance, COD
deficit, COD profiles, COD ability. While I understand the need for different terms, I suggest
unifying the terminology for consistency."
Authors’ answer: We appreciate the reviewer’s suggestion regarding terminology. However,
each term has been intentionally selected to reflect specific nuances relevant to our study:
• COD ability: refers explicitly to the player's capacity or competence to perform
directional changes effectively, viewed as a general athletic trait.
• COD performance: specifically indicates the quantified, measured outcome obtained
from tests assessing the effectiveness and speed of changing directions.
• COD deficit: clearly defined as the additional time required to perform a directional
change compared to a linear sprint over an identical distance, thus quantifying the
efficiency of directional changes relative to linear sprinting.
• COD profiles: relates to the comprehensive characterization of athletes' COD
performance and deficit across multiple tests or contexts.
Given the specific and deliberate use of these distinct yet interrelated terms, we believe
maintaining this terminology is necessary for clarity and accuracy in conveying our findings.
Nonetheless, we've ensured clear definitions for each term at their first mention, particularly
clarifying "COD deficit" (see previous comment response), to avoid reader confusion.
Comment 8: "Lines 78-79 – I fully agree with the authors, but I suggest contextualizing the
actions related to COD in futsal, such as dribbling or defensive movements, to strengthen the
connection with game dynamics."
Author’s answer: We appreciate the reviewer’s helpful feedback. The text has been enriched to
better link COD to futsal-specific actions. Revised text: “The ability to perform COD tasks, which
are crucial in futsal, notably during frequent dribbling sequences, one-on-one defensive duels,
and swift transitions between offensive and defensive roles [25]."
Comment 9: "Lines 86-88 – The authors could enhance this idea by mentioning that athletes
with higher isometric, eccentric, and concentric strength indices have greater ability to absorb
and apply propulsive force during direction changes. (Spiteri et al., 2015 – Mechanical
determinants of faster change of direction and agility performance in female basketball
athletes)."
Authors’ answer: We appreciate the reviewer's valuable suggestion, as it significantly enhances
our rationale linking muscular strength and COD performance. We have revised the text around
lines 86–88 (now on lines 99-102) as suggested, clearly incorporating the recommendation.
Revised text: "Indeed, athletes with higher levels of isometric, eccentric, and concentric
muscular strength demonstrate superior abilities to absorb and generate propulsive forces
during rapid directional changes, enhancing their COD efficiency [28]. Such muscular capacities
thus likely contribute significantly to COD performance and might influence training strategies
aimed at optimizing athletes' directional change capabilities."
Comment 10: "Lines 89-90 – Although the selected tests align with the study’s goals, the
referenced study does not include futsal athletes. I suggest reformulating the sentence or
selecting a more relevant reference."
Answer: We appreciate the reviewer's valuable feedback. As suggested, we have removed the
specific reference to futsal from the sentence, broadening its scope to accurately reflect the
cited study's context. The modified sentence now reads: "The 20m sprint and zigzag tests were
chosen in this study due to their established relevance in assessing linear and multidirectional
speed [23]."
Materials and Methods
Comment 11: "Line 119 – The authors should describe the sample in greater detail. For future
study comparisons, it is essential to specify not only the age but also the competitive level of the
athletes. While the use of Tier 2 classification is interesting, some readers may not be familiar
with it. Are all athletes from the same team? How many athletes are from U13, U15, and U17?
Information on competition level, number of participating teams, and team ranking would
enhance the description."
Authors’ answer: We appreciate the reviewer’s valuable suggestion. We appreciate the
reviewer’s valuable suggestion. We have expanded the description of the sample by clearly
including detailed age, years of practice, anthropometric and maturational characteristics, and
clarified the competitive context and composition of the participants to improve clarity and
facilitate future comparisons:
Revised text: "The participants consisted of 33 young male futsal players recruited from their
respective regional squads. They were chosen from multiple local clubs based on their superior
performance, thus forming representative regional teams. Athletes were grouped by age into
three categories: U13 (n = 10; mean age: 12.0 ± 0.55 years), U15 (n = 12; mean age: 14.2 ± 0.41
years), and U17 (n = 11; mean age: 16.2 ± 0.67 years). The players' demographics are presented
in Table 1 including the years of practice by age-group. All three age-groups had two training
blocks per week with each block lasting between one and a half and two hours. They were
evaluated immediately before their major national competition, and were considered Tier 2
athletes [33]. To be included in the measurements, players had to be completely free of pain at
the time of the study and training regularly. If someone would be receiving medical attention at
the time or indicated any pain during the Y-balance test (please report to the methods section)
they would be excluded from the study. Parents or guardians and players themselves signed an
informed consent form. All procedures were by the Declaration of Helsinki regarding human
research, and the Polytechnic Ethics Board approved the research (No. 127/2023). Ethical
Approval Date: 04-01-2023."
Comment 12: "Line 137 – Why did the authors define groups based on age rather than
maturation level?"
Authors’ answer: We appreciate the reviewer’s relevant question. Groups were defined based
on chronological age primarily due to practical and organizational constraints related to
competitive categories commonly used in futsal (U13, U15, U17). However, maturation offset
and peak height velocity (PHV) were assessed and considered within analyses, allowing us to
discuss maturation effects separately.
We have clarified this point in the manuscript as follows:
Revised text: "The subjects were split into three groups according to their chronological age
primarily due to practical and organizational constraints related to competitive categories
commonly used in futsal (U13, U15 and U17). Nevertheless, maturity offset and peak height
velocity (PHV) were assessed to account for maturational effects."
Comment 13: "Lines 163-164 – I acknowledge the challenges of data collection in team settings,
but do the authors consider that greater prior familiarization with the tests could enhance
consistency in the results?"
Authors’ answer: We appreciate the reviewer’s insightful suggestion. Indeed, while all
participants received prior familiarization with the testing procedures, additional familiarization
sessions could potentially further enhance test consistency. This information is stated in the
methods section (lines 183-186), but we added also on the Y-balance test to ensure clarity (lines
198-199).
Comment 14: "Line 165 – In Figure D, the image and caption depict a drop jump. While I
understand that the measured variable is RSI, the test name should be consistent. The RSI
calculation formula should also be provided. Additionally, the drop height (in cm) must be
specified. A detailed movement description is recommended to ensure proper execution of the
three jumps."
Authors’ answer: We appreciate the reviewer’s valuable suggestion. In the last version we didn’t
go through every detailed of each protocol to avoid overwhelming the readers. Instead, we
stated that detailed protocols could be found in the cited articles. However, as recommended,
we clarified the test names, provided the calculation formula for RSI clearly, specified the drop
height accurately as 45 cm, and provided concise but detailed descriptions of each movement,
rest periods, and execution procedures to enhance clarity and replicability.
Revised Text: "Before data collection, players performed a standardized warm-up based on
muscle activation monitored by their coach, and afterward, they became familiar with the
jumping tests’ protocols by performing each test with very little effort to understand the
biomechanics of each test and ensuring correct technique. The squat jump (SJ, measured in cm),
countermovement jump (CMJ, measured in cm), and reactive strength index (RSI, measured in
m/s) were utilized to assess lower limb strength and power. Each player performed three
attempts for each jump type, with 30 seconds of rest between attempts of the same jump type
and three minutes of rest between different jump tests to minimize fatigue [35]. For SJ, players
were instructed to jump vertically from a semi-squat position (approximately 90° knee flexion),
without countermovement [36]. For the CMJ, participants performed a rapid downward
countermovement (knee flexion to approximately 90°), immediately followed by a maximal
vertical jump [36]. For the RSI, athletes executed a drop jump from a box height of 45 cm, with
RSI calculated as the ratio between flight time (s) and ground contact time (s) [36]. Detailed
descriptions regarding test validity and reliability have been previously reported [35,36]. The
best trial was used for further analysis [36]. All tests were measured with an Optojump system
(Microgate, Bolzano, Italy) with the bars separated by 1 m [37]. The validity and reliability of this
equipment have already been confirmed [37]. Detailed protocols for each test can be found
elsewhere [35,36]."
Comment 15: "Line 188 – Court or pitch?"
Authors’ answer: We appreciate the reviewer’s observation. The text has been revised to
explicitly state "court," consistent with futsal terminology:
Revised text: " The linear sprint (partials taken at 5, 10 and 20-m) [39] and zig-zag [29] tests
were chosen as performance variables and collected on a hard surface indoor court. "
Comment 16: "Lines 186-187 – Can the authors justify the choice of tests for futsal? Why was
the 20m sprint test selected instead of 30m or 40m? Regarding COD tests, were other tests such
as the V-Cut or T-Test considered?"
Authors’ answer: We appreciate the reviewer’s valuable and thoughtful suggestion. We chose
the 20 m sprint and zigzag COD tests due to their established relevance in assessing linear and
multidirectional speed. Specifically, the 20 m sprint was selected instead of longer distances
(such as 30 m or 40 m) because futsal primarily involves short-distance sprints rarely exceeding
20 meters due to the game's restricted court dimensions and frequent interruptions. Longer
sprint distances are less representative of typical futsal match demands. Additionally, while tests
such as the V-cut or T-test were considered, the zigzag COD test was preferred due to its proven
validity, ease of implementation, and closer alignment with futsal-specific directional change
patterns. This rationale ensures alignment of the selected assessments with the practical and
physiological requirements of futsal athletes.
Revised text added: "These tests closely replicate futsal's specific physical demands,
characterized by short linear sprints, frequent accelerations and decelerations, and repeated
directional changes. Given futsal’s smaller court dimensions and frequent game interruptions,
the 20 m sprint test is more appropriate than longer distances (e.g., 30 or 40 m), ensuring
ecological validity and alignment with the sport-specific performance requirements.”
Comment 17: "Line 190 – I suggest using more specific references with greater scientific impact
to support this statement."
Author’s answer: We appreciate the reviewer’s valuable suggestion. The previously cited
reference (Nimphius et al., 2016) is indeed a highly valuable and widely recognized study
published in one of the most reputable journals in sports science, providing a rigorous
theoretical and methodological foundation for evaluating change-of-direction performance.
Nevertheless, we have further strengthened the specificity and sport-specific relevance by
additionally citing a recent futsal-specific study (Loturco et al., 2022), which explicitly examined
COD performance among elite futsal athletes.
Revised text: "The zigzag test, with its pre-planned directional changes, mimics futsal-specific
movements, while the 20-m sprint isolates linear speed, allowing the calculation of the COD
deficit – a measure of directional change efficiency [27,29].
New Reference (added):
• Loturco, I., Pereira, L.A., Reis, V.P., Abad, C.C.C., Freitas, T.T., Azevedo, P.H.S.M., &
Nimphius, S. (2022). Change of direction performance in elite players from different
team sports. Journal of Strength and Conditioning Research, 36(3), 862–866. DOI:
10.1519/JSC.0000000000003536
This approach enhances both the theoretical rigor and practical applicability of our selected
tests, addressing the reviewer’s comment effectively.
Comment 18: "Line 198 – A more detailed description of the test organization is needed to
improve replicability. Does the test involve a total linear distance of 20m? What is the distance
between markers?"
Author’s answer: We appreciate the reviewer’s request for additional clarity. The revised
description provides greater detail to ensure replicability.
Revised text: "The zig-zag test (i.e., a zigzag course with four 5-meter sections spaced at 100º
angles – 20-m total distance)”
Comment 19: "Line 212 – I recommend replacing 'by others' with 'in the literature.'"
Author’s answer: We appreciate the reviewer's suggestion and have made the recommended
adjustment.
Statistical Analysis
Comment 20: "Maturity offset and PHV can vary significantly, which may bias results when
comparing only by age. Did the authors consider using an ANCOVA to adjust age effects based
on maturation? Was homogeneity of variance tested? Was Levene’s test conducted? Authors
should specify that, in addition to Cohen’s d, 95% confidence intervals were used for effect size
precision."
Author’s answer: We sincerely appreciate the reviewer’s valuable suggestion regarding the
potential bias introduced by maturation variability. We carefully considered conducting an
ANCOVA to statistically adjust for maturation effects; however, given our limited sample size
within each age group (n between 10 and 12 per subgroup), performing an ANCOVA could
compromise statistical power and reliability due to insufficient degrees of freedom. Therefore,
we chose to clearly present age-group differences as they directly reflect practical competitive
categories in futsal, while explicitly acknowledging the potential confounding influence of
maturation offset and peak height velocity (PHV) in our discussion. Despite this methodological
limitation, we believe our analyses remain valuable and relevant by providing practical insights
aligned with current competitive categorization in youth futsal. Additionally, we confirm that
homogeneity of variance was indeed tested using Levene’s test, and we have explicitly clarified
in our manuscript that Cohen’s d was calculated with corresponding 95% confidence intervals
for precise interpretation of effect sizes.
Revised added text: "Homogeneity of variance was verified using Levene’s test. Cohen’s d was
calculated with 95% confidence intervals for precise interpretation of effect sizes."
Results
Comment 21: "I suggest reducing the written text to essential points to avoid duplicating
information between tables and text. Could Tables 2, 3, and 4 be merged?"
Author’s answer: We appreciate the reviewer’s suggestion regarding text and table
organization. Initially, these tables were presented together in a merged format, horizontally
oriented for clarity and conciseness. However, due to the journal’s template and formatting
constraints—which mandate vertical orientation—we were required to separate them into
individual tables (Tables 2, 3, and 4).
Nevertheless, we significantly revised the textual description to highlight only key findings, thus
avoiding redundancy with the tables:
Revised Results text: “Tables 2, 3, and 4 present pairwise comparisons between age groups (U13
vs. U15, U13 vs. U17, and U15 vs. U17, respectively). Anthropometric differences were observed,
with older athletes (U15 and U17) presenting significantly greater body mass and height
compared to younger athletes (U13). Regarding strength and power, significant differences
favoring older athletes (U17 over U13) were observed in CMJ, SJ, and relative power. The RSI
showed no significant differences across age groups. For dynamic balance, only the right-limb
composite score (CS right) significantly differed, with the U17 group outperforming the U13
group (mean diff. = 9.957, 95%CI = 4.16 to 15.75, p < 0.001, d = 2.59). Linear sprint performance
did not significantly differ between U15 and U17 age-groups. In the zigzag test, the U15 group
was significantly faster than the U13 group. Finally, the COD deficit was significantly lower
(better performance) in the U17 compared to the younger age groups (U13 and U15) (Tables 2–
4).”
Discussion
Comment 22: "Lines 466-469 – The authors have maturity offset and PHV data, which could
enhance discussion depth rather than just mentioning maturation."
Author’s answer: We appreciate the reviewer’s insightful suggestion. We expanded the
discussion slightly by explicitly integrating our maturity offset and PHV data to provide additional
context and depth regarding maturational influences on the COD deficit and observed
differences between age groups.
Comment 23:
"Lines 474-481 – Could the differences in COD test results between groups be related to the fact
that most U17 athletes have already passed PHV, while U15 athletes are currently experiencing
it?"
Author’s answer: We appreciate the reviewer’s valuable comment. We explicitly addressed this
possibility within the discussion by clarifying that U17 athletes had already passed their peak
height velocity (PHV), while the U15 group was experiencing or approaching PHV. We linked
these maturational differences to performance variations, particularly highlighting their possible
influence on COD test outcomes.
Language and Clarity
Comment 24: "Overall, the English language quality is good, but some phrasing could be
improved for better readability."
Author’s answer: We appreciate the reviewer’s suggestion. A thorough language revision was
conducted by a native speaker in this new version of the manuscript.

Reviewer 2 Report

Comments and Suggestions for Authors

Thank you for the opportunity to review this manuscript, which considers some interesting, applied issues.

This study appears to be novel, and author showed an interesting point about “Characterization and Comparison of Athletic Performance and COD Deficit Across Youth Futsal Age-Groups”.

The manuscript is well structured and provides significant insights into the development of COD abilities in young futsal players. However, improvements in methodology, especially regarding sampling and the use of additional tests, could further increase the reliability of your findings. Also, I believe that the sample size (n=33) is too small to draw general conclusions.

Given that the program showed a percent match: 25%, I kindly ask the authors to paraphrase all sentences that match previous published works.

Based on what I have read, I notice a few things that would be good to correct, in order to improve the quality of the article.

ABSTRACT

Add more detailed information about the sample, such as the mean age and standard deviation. Also, provide this information for each subgroup, as well as the number of participants.

INTRODUCTION

I don't think the word "Futsal" needs to start with a capital letter if it's not already at the beginning of a sentence.

Basic information about young futsal players is missing. You need to look at the skills required specifically for these age groups in the mentioned sport.

In the part of the introduction where you list your previous research, please provide numerical values. I think it will be clearer to readers to know what has been processed so far and what results have been obtained.

METHODS

Add more detailed information about the sample, such as the mean age and standard deviation. Also, provide this information for each subgroup.

I believe that using more different COD tests could better illuminate different aspects of the change in direction and allow for a more precise analysis. If the authors could include additional tests, they would provide readers with a clearer and more realistic insight based on that.

DISCUSION

In this study, a pre-planned COD test (zigzag) was used, but it would be useful to discuss reactive COD tests, which are more appropriate to real-world game conditions. In futsal, players often have to change direction reactively, which requires different neuromuscular abilities. I think the authors should look at the paper below and be aware of the fact that the results of other tests might give different conclusions, so they should not generalize. I certainly think that this segment should be mentioned in the limitations, if the authors decide not to mention this parameter.

Andrašić, S., Gušić, M., Stanković, M., Mačak, D., Bradić, A., Sporiš, G., & Trajković, N. (2021). Speed, change of direction speed and reactive agility in adolescent soccer players: Age related differences. International journal of environmental research and public health, 18(11), 5883.

REFERENCES

Approximately 70% of the references are older than 5 years, and there are even a few from the last century. I advise authors to change the references and include more recent ones, in order to provide more relevant insight into their work.

 Also, it is necessary to bold the year of publication in accordance with the instructions for authors in MDPI journals. In some places this is omitted.

Author Response

Responses to Reviewer 2 Comments:
General Comments
We sincerely appreciate Reviewer 2's valuable and insightful comments, which have greatly
improved the clarity and scientific rigor of our manuscript. Below, we provide detailed Authors’
answers and clearly state the modifications made accordingly.
Abstract
Comment 1: "Add more detailed information about the sample, such as the mean age and
standard deviation. Also, provide this information for each subgroup, as well as the number of
participants."
Authors’ answer: We appreciate the reviewer’s suggestion. We have updated the abstract to
explicitly include detailed demographic information about our sample, clearly indicating the
number of participants and the mean age with standard deviation for each subgroup (U13: n =
10, mean age = 12.0 ± 0.55 years; U15: n = 12, mean age: 14.2 ± 0.41 years; U17: n = 11, mean
age = 16.2 ± 0.67 years).
Introduction
Comment 2: "I don't think the word 'Futsal' needs to start with a capital letter if it's not already
at the beginning of a sentence."
Authors’ answer: We appreciate the reviewer’s observation. The word “The” was retained by
mistake when relocating the information to the journal’s template. Thus, the word Futsal had a
capital letter because it’s the starting word of the sentence.
Comment 3: "Basic information about young futsal players is missing. You need to look at the
skills required specifically for these age groups in the mentioned sport."
Authors’ answer: We appreciate this insightful suggestion. We revised the introduction to
include essential information regarding the specific physical and skill requirements relevant to
young futsal players. This was done explicitly by highlighting the short-distance sprints, frequent
accelerations and decelerations, abrupt changes of direction, and game-specific contexts
(offensive dribbling, defensive positioning, and transitions) as described previously.
Comment 4: "In the part of the introduction where you list your previous research, please
provide numerical values. I think it will be clearer to readers to know what has been processed
so far and what results have been obtained."
Authors’ answer: We sincerely appreciate the reviewer’s suggestion. We have carefully revised
the introduction to explicitly incorporate numerical values from previous research, which
enhances the clarity of the background and strengthens the justification for our study.
Revisions Made:
• Futsal Physical Demands: We added values related to high-intensity sprinting efforts,
specifying that elite futsal players engage in an average of 22 sprints per match and
spend 25–35% of total match time at speeds exceeding 18 km/h [7]. Additionally, semiprofessional players typically perform sprints lasting 2–4 seconds, reinforcing the
importance of sprint and change-of-direction ability in futsal [8].
• Performance Differences Between Competitive Levels: In discussing previous research
on physical performance differences in futsal players, we included specific numerical
data from Ayarra et al. (2022). This revision clarifies that despite expectations, no
significant differences in COD test times were observed between Second Division B (5.41
± 0.26 s) and Third Division (5.48 ± 0.32 s; p = 0.68). Additionally, countermovement
jump (CMJ) values were comparable across groups (Second Division B: 43.6 ± 5.6 cm;
Third Division: 45.8 ± 4.3 cm; Junior: 43.4 ± 4.1 cm), reinforcing that physical
performance stabilizes once players reach adulthood [2].
These additions ensure that previous research is presented with greater precision, fully
addressing the reviewer’s concern while maintaining a clear and concise introduction.
Methods
Comment 5: "Add more detailed information about the sample, such as the mean age and
standard deviation. Also, provide this information for each subgroup."
Authors’ answer: We appreciate the reviewer’s suggestion. We have carefully expanded the
description of the participants' characteristics to include mean age, standard deviations, and
clarified their competitive level as regional representatives from multiple local clubs. This
information is explicitly included in the revised Table 1 and in the text description of the sample.
Comment 6: "I believe that using more different COD tests could better illuminate different
aspects of the change in direction and allow for a more precise analysis. If the authors could
include additional tests, they would provide readers with a clearer and more realistic insight
based on that."
Authors’ answer: We appreciate this insightful suggestion. We acknowledge that employing
multiple COD tests could indeed provide a broader perspective. However, given the practical
constraints (e.g., total testing time available, the risk of fatigue, and logistical issues), we
carefully selected the zigzag COD test based on its established validity and practicality in futsal
contexts. This limitation has now been clearly stated in our discussion, and we explicitly suggest
that future studies might incorporate additional COD assessments, particularly reactive COD
tests, to further enhance insight into real-game demands.
Discussion
Comment 7: "In this study, a pre-planned COD test (zigzag) was used, but it would be useful to
discuss reactive COD tests, which are more appropriate to real-world game conditions. In futsal,
players often have to change direction reactively, which requires different neuromuscular
abilities. I think the authors should look at the paper by Andrašić et al. (2021), and be aware of
the fact that the results of other tests might give different conclusions, so they should not
generalize. I certainly think that this segment should be mentioned in the limitations if the
authors decide not to mention this parameter."
Authors’ answer: We sincerely appreciate the reviewer’s important observation. Although the
zigzag test effectively assesses pre-planned COD performance, reactive COD tests would indeed
offer more ecologically valid insights into futsal-specific performance. We explicitly addressed
this limitation in the revised discussion, acknowledging that using reactive agility tests could
potentially yield different results and emphasizing that generalization should be made
cautiously. This was added explicitly in the limitations section of our manuscript. We did take a
look into the paper by Andrašić et al. (2021), but we decided not to cite it there because the
paper is related to soccer.
Revised Discussion/Limitations Text (example): "(…) Futsal requires rapid reactive changes of
direction involving distinct neuromuscular mechanisms, which were not directly assessed in this
study. Therefore, future research should incorporate reactive COD tests to obtain deeper
insights into futsal-specific performance demands and to ensure broader applicability of the
results"
References
Comment 8: "Approximately 70% of the references are older than 5 years, and there are even a
few from the last century. I advise authors to change the references and include more recent
ones, in order to provide more relevant insight into their work. Also, it is necessary to bold the
year of publication in accordance with the instructions for authors in MDPI journals. In some
places this is omitted."
Authors’ answer: We sincerely appreciate the reviewer’s observation and valuable suggestion.
We acknowledge the importance of providing the most current literature to ensure relevance
and context. Following your suggestion, we have carefully reviewed our reference and made our
effort to include additional recent publications.
However, we would also like to respectfully highlight that some older references in our
manuscript are foundational sources providing well-established definitions, methodological
descriptions, or widely accepted protocols within our research domain. Such references, despite
their publication date, remain staples and continue to hold significant value and relevance
within the scientific community. For example, some methodological and conceptual papers
remain authoritative references, widely recognized and frequently cited in recent publications.
We kindly hope that you appreciate our careful effort to maintain a balanced approach between
foundational literature and more contemporary studies. Additionally, we have formatted our
references by clearly bolding the publication years, adhering strictly to MDPI guidelines.
Regarding plagiarism concern:
Comment 9: "Given that the program showed a percent match: 25%, I kindly ask the authors to
paraphrase all sentences that match previously published works."
Authors’ answer: We sincerely appreciate your concern. We carefully reviewed and
paraphrased all sentences throughout the manuscript.
Overall Sample size comment:
Comment 10: "I believe that the sample size (n=33) is too small to draw general conclusions."
Authors’ answer: We appreciate the reviewer’s thoughtful observation. We fully acknowledge
this limitation regarding sample size explicitly within our manuscript. Given our study’s
preliminary nature and the practical challenges associated with recruiting large samples of
competitive youth futsal athletes, our findings should be interpreted cautiously and primarily
considered within the context of the studied population rather than generalized broadly. Also,
it is important to point out that this sample represented the best futsal players of the district.
Only the selected best players from all regional clubs were included. Thus, it would be
challenging to expand the sample in this specific study.

Round 2

Reviewer 1 Report

Comments and Suggestions for Authors

I would like to commend the authors for their valuable work and acknowledge their efforts in enhancing the quality of the manuscript. Their dedication to refining the content is evident, and the improvements significantly strengthen the overall clarity and impact of the study.